# The β1 Adrenergic Blocker Nebivolol Ameliorates Development of Endotoxic Acute Lung Injury

**DOI:** 10.3390/jcm12051721

**Published:** 2023-02-21

**Authors:** Esra Nurlu Temel, Mehtap Savran, Yalcın Erzurumlu, Nursel Hasseyid, Halil Ibrahim Buyukbayram, Gozde Okuyucu, Mehmet Abdulkadir Sevuk, Ozlem Ozmen, Ayse Coskun Beyan

**Affiliations:** 1Department of Infectious Diseases, Faculty of Medicine, Suleyman Demirel University, 32260 Isparta, Turkey; 2Department of Pharmacology, Faculty of Medicine, Suleyman Demirel University, 32260 Isparta, Turkey; 3Department of Biochemistry, Faculty of Pharmacy, Suleyman Demirel University, 32260 Isparta, Turkey; 4Department of Medical Biochemistry, Faculty of Medicine, Suleyman Demirel University, 32260 Isparta, Turkey; 5Department of Pathology, Faculty of Veterinary Medicine, Burdur Mehmet Akif Ersoy University, 15030 Burdur, Turkey; 6Department of Occupational Medicine, Faculty of Medicine, Dokuz Eylul University, 35220 İzmir, Turkey

**Keywords:** apoptosis, inflammation, lipopolysaccharides, nebivolol, oxidative stress, respiratory system

## Abstract

Acute lung injury (ALI) is a disease, with no effective treatment, which might result in death. Formations of excessive inflammation and oxidative stress are responsible for the pathophysiology of ALI. Nebivolol (NBL), a third-generation selective β1 adrenoceptor antagonist, has protective pharmacological properties, such as anti-inflammatory, anti-apoptotic, and antioxidant functions. Consequently, we sought to assess the efficacy of NBL on a lipopolysaccharide (LPS)-induced ALI model via intercellular adhesion molecule-1 (ICAM-1) expression and the tissue inhibitor of metalloproteinases-1 (TIMP-1)/matrix metalloproteinases-2 (MMP-2) signaling. Thirty-two rats were split into four categories: control, LPS (5 mg/kg, intraperitoneally [IP], single dose), LPS (5 mg/kg, IP, one dosage 30 min after last NBL treatment), + NBL (10 mg/kg oral gavage for three days), and NBL (10 mg/kg oral gavage for three days). Six hours after the administration of LPS, the lung tissues of the rats were removed for histopathological, biochemical, gene expression, and immunohistochemical analyses. Oxidative stress markers such as total oxidant status and oxidative stress index levels, leukocyte transendothelial migration markers such as MMP-2, TIMP-1, and ICAM-1 expressions in the case of inflammation, and caspase-3 as an apoptotic marker, significantly increased in the LPS group. NBL therapy reversed all these changes. The results of this study suggest that NBL has utility as a potential therapeutic agent to dampen inflammation in other lung and tissue injury models

## 1. Introduction

Sepsis is an exaggerated response that a host develops against an infection; the lungs are greatly affected because they have excessive blood flow. More than half of patients diagnosed with sepsis experience lung problems, such as acute lung injury (ALI). Despite current antibiotic and supportive treatment strategies to combat sepsis and related pulmonary problems, ALI still presents a notable mortality rate [1,2,3].

In a typical ALI, extensive pulmonary inflammation, which results in the apoptosis of alveolar epithelial cells, is the main mechanism of damage. The inflammatory state of the lungs can be modeled by a challenge with LPS, a part of the Gram-negative bacterial cell wall, to understand the underlying mechanisms of ALI [4]. The recognition of LPS by a specific Toll-like-receptor-4 (TLR-4) receptor on the pulmonary epithelium stimulates the secretion of pro-inflammatory cytokines such as tumor necrosis factor-alpha (TNF-α), which subsequently increases inflammation and capillary membrane permeability. Increased permeability directs leukocytes to interstitial lung tissue. In addition, inflammation is known to induce oxidative stress, DNA damage, apoptosis, and necrosis, all of which contribute to the ALI process individually [4,5,6,7,8].

Metalloproteinases (MMP), whose zymogen forms originate from various types of cells and are secreted into the matrix, are proteinases that regulate the tissue structure by degrading extracellular matrix (ECM) components [9]. The level of MMP-2 decreases after pulmonary maturation, whereas its expression is induced in inflammation to create a chemotactic gradient to eliminate inflammatory cells [10].

TIMP-1 is a member of the protein family that functions as a tissue inhibitor of MMPs, especially MMP-2 and MMP-9 [11]. The balance of MMPs/TIMP-1 is important for ECM turnover in the lung. Additionally, TIMP-1 is known to be induced by acute-phase reactants in inflammation, so the balance of MMPs/TIMP-1 is affected by inflammatory cell secretions [12,13].

ICAM-1 is a cell-surface glycoprotein expressed in the endothelium and epithelium. It regulates leukocyte recruitment from the circulation to infections due to induction by inflammatory cytokines [14].

NBL, a third-generation β1-adrenoreceptor antagonist, acts as a vasodilator agent by enhancing the levels of nitric oxide (NO) in addition to the other properties of beta blockers. In the literature, NBL has been shown to modulate the inflammation both in vivo and in vitro. In addition to reducing the inflammation that occurred in hypertensive patients, NBL protected the lung, liver, kidney of rats and type-II collagen tissue in cell cultures by reducing various cytokines of inflammation, such as IL-6, ICAM-1, and TNF-α [15,16,17,18,19,20,21,22].

Despite research on the protective effects of NBL in cardiovascular pathologies, the beneficial features of NBL on ALI are not yet known. Therefore, this research is the first to investigate NBL’s impact on inflammation-induced lung injury through transendothelial migration via MMP2/TIMP-1 signaling and ICAM-1 expressions.

## 2. Materials and Methods

Suleyman Demirel University’s local animal ethics commission approved this research (Decision Number 06-25, 11.09.20). The experiment was conducted according to ARRIVE (Animal Research: Reporting in Vivo Experiments) guidelines, Version 2.0 protocol. Thirty-two adult male Wistar albino rats, weighing 300–350 g, were housed at 21–22 °C and 60 ± 5% humidity with a 12 h light:12 h dark cycle. A standard commercial chow (Korkuteli Yem, Antalya, Turkey) was administered ad libitum with water. Four experimental groups were formed as follows:

Negative controls (n = 8): For three days, 1 mL of normal saline (NS) was administered via oral gavage. Thirty minutes after the last NS, one dosage of 1 mL of NS was injected IP into the right inguinal region of the rat.

Positive (LPS) controls (n = 8): For three days, 1 mL of NS was administered via oral gavage. Thirty minutes after the last NS, one dosage of 5 mg/kg LPS (048K4126, Sigma Aldrich, USA) soluble in NS was injected IP into the right inguinal region of the rat [23].

LPS + NBL group (n = 8): For three days, 1 mL of 10 mg/kg NBL (Nexivol, Abdi İbrahim, Turkey) soluble in NS was applied via oral gavage. Thirty minutes after the last NBL administration, one dosage of 5 mg/kg LPS soluble in NS was injected IP into the right inguinal region [24].

NBL group (n = 8): For three days, 1 mL of 10 mg/kg NBL soluble in NS was administered via oral gavage, and one dosage of 1 mL of IP saline was applied 30 min after the last NBL administration.

For anesthesia, 100 mg/kg ketamine (Alfamine, Alfasan IBV) and xylazine 10 mg/kg (Alfazin, Alfasan IBV) were administered 6 h after LPS stimulation [24]. Surgical exsanguination by blood withdrawal from the inferior vena cava following abdominal incision was used for euthanasia. After lung tissue extraction for histological evaluation, they were inflated and frozen in liquid nitrogen and kept at −20 °C for gene expression and biochemical examination. The remaining extracted tissues were maintained for immunohistochemical and histopathological examination following fixation with 10% buffered formalin.

### 2.1. Histopathological Analyses

The routine pathology tissue-processing of pulmonary specimens was carried out by an automatic tissue processor (Leica-ASP300S, Wetzlar, Germany), followed by embedding in paraffin wax. Afterward, 5μm-thick pieces were cut from the paraffin blocks using a Leica-RM2155 rotary microtome (Leica Microsystems, Wetzlar, Germany). Subsequently, hematoxylin-eosin (H&E) staining was performed by mounting with a coverslip and examination under a light microscope. For the histopathological injury score of the lungs, each rat was evaluated by two expert pathologists who were unaware of the groups using a modified scoring system [25]. Histopathological lesions were graded from 0 to 3, as shown in Table 1, by evaluating the findings of hyperemia, edema, infiltration, and septal thickening. By averaging the evaluations of the two pathologists, a score for each animal was determined.

### 2.2. Immunohistochemical Analyses

For the immunohistochemistry, two series of sections from all blocks drawn on poly-L-lysine-coated slides were stained for the expression of caspase-3 (Cas-3) [Anti-caspase-3 Antibody (E-8):sc-7272, 1/100 dilution] and ICAM-1 [ICAM-1(M/K-2):sc-18864, 1/100 dilution, Santa Cruz, Texas, USA] using the streptavidin–biotin method as per the manufacturer’s instructions. For 60 min, the section was incubated with the primary antibody. Secondary antibodies (EXPOSE Mouse and Rabbit Specific HRP/DAB Detection IHC kit [ab80436, Abcam, Cambridge, UK]) included streptavidin-alkaline phosphatase conjugate and biotinylated secondary antibodies. Diaminobenzidine (DAB) was chosen as the chromogen (DAB). An antigen dilution solution, but not the primary antibody, was used for negative controls. Two specialized pathologists from different universities evaluated all the blinded samples. The sections were independently examined for individual antibodies during immunohistochemical analyses. Semiquantitative analyses were performed to assess the immunostaining degree of cells using a ranking from 0 to 3: 0 = no staining, 1 = poor focal staining, 2 = poor diffuse staining, and 3 = strong diffuse staining [26].

For each section, 10 fields at 400X magnification were evaluated. Statistical analyses were conducted by averaging the scores of both pathologists for each lung. The Database Manual cellSens Life Science Imaging Software System (Olympus Co., Tokyo, Japan) was used for microphotography and morphometric analysis.

### 2.3. Biochemical Analyses

First, the lung tissues were diluted five-fold (*w*/*v*) with phosphate-buffered saline (10 mM sodium phosphate) at pH 7.4. Next, tissues were homogenized with a tissue homogenizer (IKA Ultra Turrax T25, Janke & Kunkel, Staufen, Germany). At the end of the homogenization, the samples were centrifuged at 2000 rpm/20 min./+4 °C (Nüve NF 1200R, Ankara, Turkey). Tissue total antioxidant status (TAS) and total oxidant status (TOS) concentrations were assayed from the supernatant of the samples using an automated biochemistry analyzer (Beckman Coulter AU 5800, Brea, CA, USA) and colorimetric methods developed by Erel [27,28,29].

TOS results were μmol H_2_O_2_ Equiv./L, and TAS results were mmol Trolox Equiv/l. The oxidative stress index (OSI) was calculated by dividing the TOS levels by TAS levels, that is, TOS/TAS × 100 [25].

### 2.4. Quantitative Polymerase Chain Reaction Analysis

RNA was isolated from lung tissues using TRIzol^TM^ (Thermo Fisher Scientific, Carlsbad, CA, USA) (New England BioLabs). The amount and purity of the RNAs were measured using a nanodrop device (VWR MySPEC, Darmstadt, Germany). Moreover, 1 μg of RNA of 1.8–2.1 purity was taken, and cDNA was obtained in a thermal cycler (Thermo Scientific, Waltham, MA, USA) with the iScript cDNA Synthesis kit according to the manufacturer’s protocol. Real-time PCR amplification was performed with a Bio-Rad CFX96 instrument using the iTaq Universal SYBR Green Supermix (Bio-Rad, Hercules, CA, USA). Primers were designed using the NCBI primer-BLAST (Table 2). GAPDH expression was used for normalization. The reaction volume was adjusted to 25 μL using 100 ng of the cDNA sample, and the RT–qPCR conditions were determined according to the manufacturer’s protocol. The relative measurement of gene expression was executed using the Livak method and the 2^−ΔΔCt^ method [30]. A melting curve analysis was performed on RT–qPCR products to ascertain the specificity of the amplification. Fold changes are given graphically (Table 2).

### 2.5. Statistical Analyses

For statistical analyses of the immunohistochemical and biochemical scores of the groups, the SPSS 22.00 (SPSS Inc., Chicago, IL, USA) program was used, and the significance of the scores were compared between the groups. One-way ANOVA and post hoc Bonferroni and Duncan tests were conducted to evaluate the significance among the groups. The data are presented as the mean ± SD. Values with *p* < 0.05 were deemed significant.

## 3. Results

### 3.1. Immunohistochemical and Histopathological Results

In the control group, the histopathological investigation indicated a normal histoarchitecture. Severe hyperemia, interstitial edema, and increased thickness of septal tissue were the marked findings in the LPS group. In addition, inflammatory cell infiltrations mainly comprising neutrophils were commonly seen in this group. NBL treatment decreased histopathological findings. Normal lung histology was observed in the NBL group (Figure 1). In addition, the degree of lung damage was scored using a semiquantitative histopathology system according to alveolar septa hyperemia, alveolar hemorrhage, edema, and neutrophil leukocyte infiltrations.

It was observed that LPS resulted in a substantial rise in histopathological scores in the lungs. However, NBL treatment markedly lessened lung injury scores (Table 3). In an immunohistochemical examination, the negative to slight expression of Cas-3 and ICAM-1 was seen in the control group. Comparing the LPS group to the control, there was a significantly higher expression of Cas-3 and ICAM-1 (*p* < 0.001). NBL treatment was associated with a marked decrement in these expressions compared with animals subjected to LPS (*p* < 0.001). See Figure 2 and Figure 3.

Only in the NBL-administered group were negative to very slight expressions noticed. Immunohistochemical expressions were detected in lung alveolar cells and in inflammatory cells. The results of the statistical analyses of immunohistochemical scores are displayed in Table 3.

### 3.2. Biochemical Results

TAS levels did not differ significantly across groups. TOS levels considerably increased in the LPS (*p* < 0.001) and LPS + NBL groups (*p* < 0.01) compared to the controls. TOS levels were markedly lower in the LPS + NBL group compared to the LPS exposure (*p* < 0.001). In only the NBL-administered group, TOS values were notably lower compared to the LPS and LPS + NBL groups (*p* < 0.001 and *p* < 0.05, respectively).

OSI levels were slightly higher in the LPS (*p* < 0.001) and LPS + NBL groups (*p* < 0.01) compared to the negative controls. OSI levels were noticeably lower (*p* < 0.01) in LPS plus NBL-exposed animals compared to only LPS exposure. In only the NBL-administered group, OSI values were considerably lower compared to the LPS and LPS + NBL groups (*p* < 0.001 and *p* < 0.05, respectively). See Figure 4.

### 3.3. Gene Expression Results

The TIMP-1 level was noticeably higher in the LPS group than in the control, LPS + NBL, and NBL groups (*p* < 0.001 for all). TIMP-1 levels were higher in the LPS + NBL group than in the control (*p* = 0.019) and NBL groups (*p* < 0.001). In the NBL-administered group, TIMP-1 levels were considerably lower than in the control group (*p* < 0.001).

MMP-2 levels were noticeable higher in the LPS group than in the control, LPS + NBL, and NBL groups (*p* < 0.001 for all). In addition, MMP-2 levels were higher in the LPS + NBL group than in the control and NBL groups (*p* = 0.042 and *p* < 0.001, respectively). MMP-2 levels were considerably lower only in the NBL-administered group than in the control (*p* < 0.001). See Figure 5.

## 4. Discussion

This study, using a chemically induced ALI model, showed the presence of increased markers of inflammation (ICAM-1, MMP-2 and TIMP-1), oxidative stress (TOS and OSI) and apoptosis (Cas-3) in the LPS groups. NBL treatment restored these alterations.

Many harmful factors, such as infection, activate receptors on cell-surface membranes. In addition, this triggers some post-receptor intracellular signaling mechanisms and leads to the production of some inflammatory cytokines such as TNF-α, interleukin (IL) 1-beta, and IL-6 [31]. These cytokines aggravate leucocyte transendothelial migration by enhancing capillary membrane permeability. It was also shown that the secretion and activation of nuclear factor kappa beta (NF-kB) is the primary stone for inducing the abovementioned proinflammatory cytokines. NBL has been known to mitigate NF-kB activation and subsequent proinflammatory cytokine secretion [22]. Thus, NBL inhibits transendothelial migration by reducing capillary membrane permeability mediated by secreted cytokines.

The damage findings observed in the groups treated with LPS indicate inflammation, which supports the literature [32]. Inflammatory cell infiltration with a predominance of neutrophils indicates pulmonary damage begins to develop in the early phase of inflammation as an acute response. Neutrophils play a pivotal role in ALI, as their levels in various body fluids relate to the severity of diseases [33]. However, neutrophilic infiltration into an injury site develops either in an integrin-dependent or independent manner, varying according to the type of inflammatory stimulus [34]. Although there is a paper that shows integrin-independent neutrophilic migration in an LPS-challenged lung model [35], it has been commonly stated that in LPS-induced models, integrin proteins, such as ICAM-1, are involved in neutrophil migration in pulmonary tissue [36]. In our study, increased expressions of ICAM-1 in LPS-treated groups emphasized that ICAM-1 is a critical component of inflammation and may explain the necessity of ICAM-1 for the transmigration of neutrophils. However, the effect of NBL on ICAM-1 expression in cardiac tissues has been demonstrated in some in vivo and in vitro studies [16,37]. On the other hand, our study is the first to illustrate the decreasing effect of NBL on pulmonary ICAM-1 level. Except for ICAM-1, other adhesion molecules such as ICAM-2, ICAM-3, E and P selectins, and vascular cell adhesion molecule-1 (VCAM-1) have also been indicated to decrease by NBL [18,38]. These molecules could be downregulated in lung tissues treated by NBL and should be evaluated in further studies.

Edema due to greater capillary permeability is one of the most characteristic features of ALI and eventually impairs oxygenation. Edema also accompanies neutrophilic infiltration during lung injury [33]. In addition, the septal thickening of alveolar membranes contributes to damage. To combat impaired oxygenation and related inflammation, enhanced expression of hypoxic inducible factor alpha (Hıf-1α) triggers eNOS-related NO production [39]. In a recent study, NBL has been shown to induce AKT1/eNOS/HIF1 α signaling to decrease inflammatory conditions [20]. In our study, edema and septal thickening in alveolar membranes increased due to inflammation caused by LPS. The observed decrements in the findings due to NBL confirm its anti-inflammatory properties. Enhancements in these parameters are important, as they can directly reflect the clinical response by recovering breathing.

The roles of MMPs and the proteolytic degradation of ECM in physiological and pathological processes are well known [40]. Moreover, proteolytic reactions can activate or inactivate inflammatory cytokines and chemokines and antagonize chemokine functions to regulate the inflammatory process [41]. Studies have shown that raising MMP-2 levels with LPS administration causes pulmonary ECM degradation and aggravates pulmonary inflammation [42,43,44,45]. MMP-2-induced degradation products can exacerbate inflammation, so MMP-2 has become a critical marker for acute inflammatory lung injuries [46]. To the best of our knowledge, LPS-induced elevations in MMP-2 levels were noticed in our study. Papers that focus on the effect of NBL on MMP-2 levels have been performed on cardiac and vascular tissues, as NBL is indicated in hypertension. However, these studies reveal conflicting results [47,48,49,50]. Furthermore, MMP-2 levels, shown for the first time to decrease with NBL in lung tissue in our study, are crucial for detailing the effect of NBL on acute lung pathologies. One of the critical results revealed by our research is the decrease in MMP-2 levels observed in the NBL-only treated group. Although there are publications reporting NBL-induced decreases in MMP-2, these studies were performed on hypertensive rats with an inflammatory condition [47,48]. The findings in our study show that NBL has the potential to decrease MMP-2 levels even without a provocative challenge in lung tissue, and this is an important result for further studies characterizing anti-inflammatory mechanisms.

TIMPs regulate the activity of MMPs [51]. Therefore, TIMP-1, is linked with various inflammation-based diseases. Although TIMP-1 participates in post-injury tissue repair processes and the resolution of inflammation in an anti-inflammatory condition, it acts as a pulmonary macrophage-derived proinflammatory cytokine in the lung to strengthen acute inflammation [52]. In the current study, the elevation of TIMP-1 should be considered an acute marker of inflammation rather than MMP-2 inhibition. Thus, simultaneous increases in TIMP-1 and MMP-2 in the LPS group and ameliorations in these values by NBL exhibit the inflammatory and anti-inflammatory natures of LPS and NBL, respectively. Transcriptional changes seen in TIMP-1 and MMP-2 could be considered moderate due to the acute design of the study. However, these alterations could be more dramatic later and should be clarified in further studies. The effects of NBL on TIMP-1 has been studied in cardiac tissue. Although no effect on TIMP-1 levels by NBL was shown in some studies [47,49], others reported decreased TIMP-1 levels by NBL [38]. As our study demonstrated, lower MMP-2 and TIMP-1 levels by NBL may reflect tissue remodeling in the lung, which revealed improvements in alveolar septal thickness using histological analysis. Additionally, in a renal fibrosis model, ICAM-1 was upregulated as one of the substrates of TIMP-1 [53]. In our study, it also can be said that TIMP-1 levels modulate inflammation over ICAM-1 expression and, hence, neutrophil transmigration. Similar to MMP-2 results, TIMP-1 levels, too, were decreased in the NBL-only treated group in addition to LPS + NBL. Although TIMP-1 levels were shown to be decreased in a cell-culture model and in senile heart failure, our result is the first to show that NBL could decrease TIMP-1 in the absence of inflammation, which may indicate the anti-inflammatory potential of NBL [49,54]

During the development of ALI, the influx of neutrophils and macrophages into the alveolar space causes the secretion of reactive oxygen species (ROS) and other proinflammatory cytokines. Although it is physiologically involved in cellular functions, excessive oxidative stress, similar to inflammation, causes tissue damage [55]. LPS-induced oxidative damage is a well-known consequence and is also depicted in our study [56]. Nonetheless, the antioxidant features of NBL have been shown in cardiac tissue [57,58]. In our research, the protective effects of NBL against oxidative stress were indicated by decreased TOS and OSI levels, without changes in TAS levels. These results suggest that NBL reduced oxidative stress secondary to the regression of inflammation, not by increasing the amount of antioxidant enzymes.

LPS induces epithelial and endothelial apoptosis in the lung, and preexisting inflammation and oxidative stress also help develop apoptosis [59]. In the current study, histopathologically, a significant increase in LPS and reduction in the NBL-treated group were observed using cas-3. Considering that the increase in cas-3 is associated with cell death, NBL can be said to illustrate an anti-apoptotic effect similar to that in the literature [60].

Briefly, NBL suppressed inflammation, oxidative stress, and apoptosis in LPS-induced ALI and prevented the extravasation of leukocytes by decreasing ICAM-1, TIMP-1, and MMP-2 expression (see Figure 6).

Although considerable scientific progress has been achieved in epidemiology to treat ALI, these advances are inadequate for reducing morbidity and mortality. As seen during COVID-19, ALI remains an unresolved, serious medical problem. The possibility that NBL, mainly used in cardiovascular pathologies such as hypertension, may help expand lung tissue by preventing leukocyte migration seems to be very attractive information. The reflection and translation of these preclinical results into clinical practice will help in the fight against inflammation-based diseases.

## Figures and Tables

**Figure 1 jcm-12-01721-f001:**
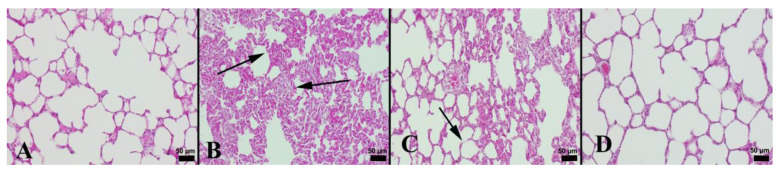
Histopathological appearance of the lungs in study groups: (**A**) normal microscopical appearance in lungs in a rat belonging control group; (**B**) marked increase in septal tissue thickness (arrows) in lungs in a rat from LPS group; (**C**) marked amelioration and decreased septal tissue thickness (arrow) in a rat in LPS + NBL group; and (**D**) normal lung tissue architecture in a rat belonging NBL group, H&E, scale bars = 50 µm.

**Figure 2 jcm-12-01721-f002:**
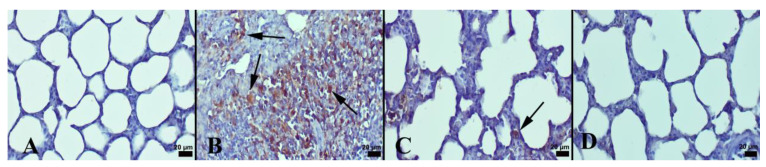
Cas-3 immunohistochemical findings among the groups: (**A**) negative expression in control group; (**B**) increased Cas-3 expression (arrows) in both alveolar epithelial cells and inflammatory cells (arrows) in LPS group; (**C**) decreased expression (arrow) in LPS + NBL group; and (**D**) no expression in NBL group, Streptavidin biotin method, scale bars = 20 µm.

**Figure 3 jcm-12-01721-f003:**
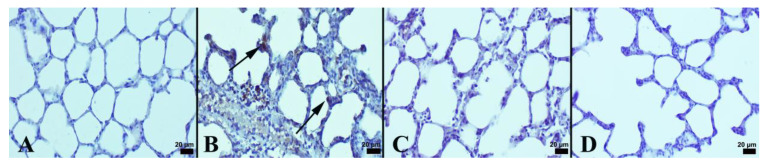
ICAM-1 expressions between the groups: (**A**) negative expression in control group; (**B**) increased expression (arrows) in alveolar cells (arrows) in LPS group; (**C**) decreased expression (arrow) in LPS + NBL group; and (**D**) negative expression in NBL group, Streptavidin biotin method, scale bars = 20 µm.

**Figure 4 jcm-12-01721-f004:**
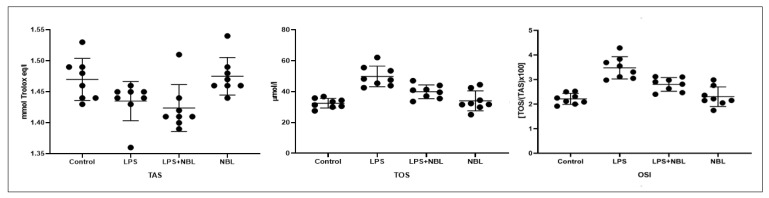
TAS, TOS, and OSI Levels in Lung Tissue. LPS: Lipopolysaccharide, NBL: Nebivolol, OSI: Oxidative Stress Index, TAS: Total Antioxidant Status, TOS: Total Oxidant Status.

**Figure 5 jcm-12-01721-f005:**
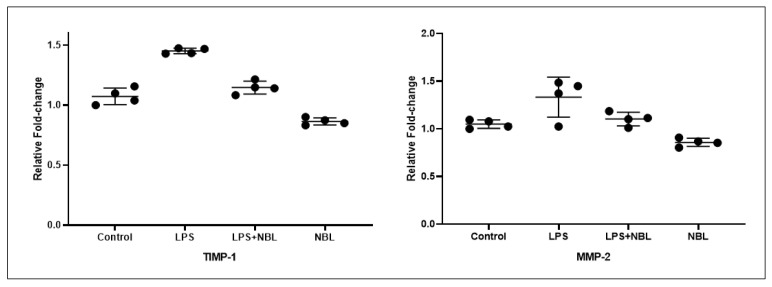
TIMP-1 and MMP-2 Levels in Lung Tissue. LPS: Lipopolysaccharide; NBL: Nebivolol; TIMP-1: TIMP Metallopeptidase Inhibitor 1; MMP-2: Matrix Metalloproteinase-2. Expression data were calculated relative to saline control.

**Figure 6 jcm-12-01721-f006:**
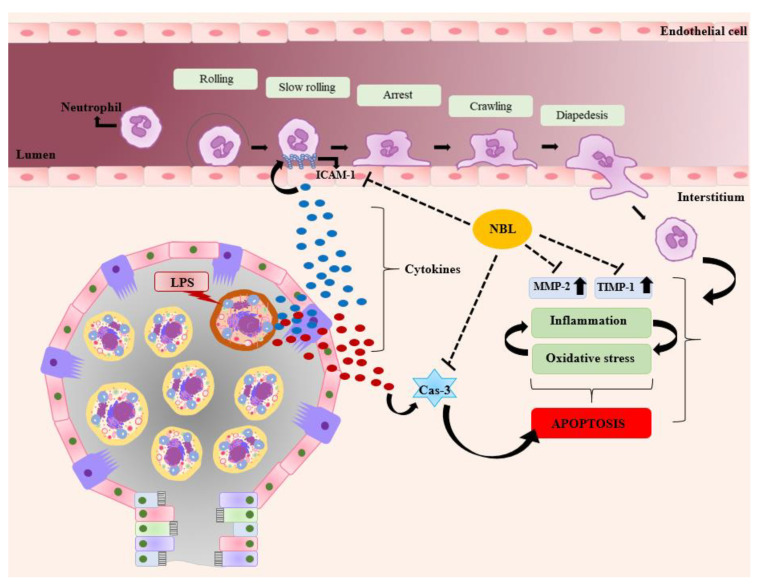
The possible effect of NBL on the pathophysiological mechanisms of LPS-induced lung injury; Cas-3: Caspase-3, ICAM-1: Intercellular Adhesion Molecule-1, LPS: Lipopolysaccharide, MMP-2: Matrix Metallo-proteinase-2, TIMP-1: Tissue inhibitor of metalloproteinase NBL: Nebivolol.

**Table 1 jcm-12-01721-t001:** Histopathological score criterion for lungs.

Score	Alveolar Septae Hyperemia	Alveolar Edema	Alveolar Hemorrhage	Intra-Alveolar Neutrophil Filtrations per Field
0	Normal thin septae	No edema	No hemorrhage	Less than 5 in the fields
1	Slight hyperemic alveolar septae (in less than 1/3 of the fields)	Slight edema (in less than 1/3 of the fields)	Less than 10 erythrocytes in the fields	5 to 10 in the fields
2	Moderate hyperemic alveolar septae (in 1/3 to 2/3 of the fields)	Moderate edema (in 1/3 to 2/3 of the fields)	11 -20 erythrocytes in the fields	10 to 20 in the fields
3	Severe hyperemic alveolar septae (in greater than 2/3 of the fields)	Severe edema (in greater than 2/3 of the fields)	More than 21 erythrocytes in the fields	More than 20 in the fields

Examinations done high-power field (HPF) (10 fields of view at ×400 magnification).

**Table 2 jcm-12-01721-t002:** Primary sequences of genes and melting temperature (Tm) values.

Gene	Specific Primer Sequence	Tm
GAPDH (HouseKeeping)	F: 5′-CAAGGTCATCCCAGAGCTGAA-3′	62.8 °C
	R: 5′-CATGTAGGCCATGAGGTCCAC-3′	
TIMP-1	F: 5′-ATGTCCACAAGTCCCAGAAC-3′	57.6 °C
	R: 5′-AGCAGGGCTCAGATTATGC-3′	
MMP-2	F: 5′-CTGTCCCGACCAAGGATATAG-3′	57.0 °C
	R: 5′-CTTGGTGTAGGTGTAGATAGGG-3	

GAPDH: Glucose 6 Phosphate Dehydrogenase, TIMP-1: Metallopeptidase Inhibitor 1, MMP-2: Matrix metalloproteinase-2, Tm: melting temperature.

**Table 3 jcm-12-01721-t003:** Results of statistical analysis of immunohistochemical scores between the groups.

Groups	Histopathological Scores	Cas-3 Scores	ICAM-1 Scores
Control	0.25 ± 0.16 ***^, ###^	0.25 ± 0.16 ***	0.87 ± 0.35 ***^, ###^
LPS	2.62 ± 0.51	1.62 ± 0.51	2.75 ± 0.46
LPS + NBL	1.25 ± 0.46 ***	0.62 ± 0.51 ***	1.37 ± 0.51 ***
NBL	0.12 ± 0.12 ***^, ###^	0.12 ± 0.12 ***	0.62 ± 0.51 ***^, ###^

Cas-3: Caspase-3, ICAM-1: Intercellular adhesion molecule-1, LPS: Lipopolysaccharide, NBL: Nebivolol. ***; *p* < 0.001, LPS and others. ###; *p* < 0001, LPS + NBL and others. Data expressed mean ± standard deviation (SD). One-way ANOVA Duncan test.

## Data Availability

The data presented in this study are available on request from the corresponding authors.

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
