# Peer review of "The β1 Adrenergic Blocker Nebivolol Ameliorates Development of Endotoxic Acute Lung Injury"

_jcm, 2023, doi:10.3390/jcm12051721_

Round 1
Reviewer 1 Report
Esra Nurlu Teme et al address an important and clinically relevant question on the effects of Nebivolol (NBL) on acute lung injury. The authors utilized a lipopolysaccharide (LPS)-induced ALI model. They aimed to evaluate the efficacy of the NBL on the ALI by the modulating of intercellular adhesion molecule-1 (ICAM-1) expression and the tissue inhibitor of metalloproteinases-1(TIMP-1)/matrix metalloproteinases-2 (MMP-2) signaling. They then considered Oxidative stress markers such as total oxidant status and oxidative stress index levels, leukocyte transendothelial migration markers such as MMP-2, TIMP-1, and ICAM-1 expression to assess inflammation, and caspase-3 as an apoptotic marker. They found that these markers significantly increased in the LPS group and NBL therapy reversed all these changes. Overall, this project addresses an interesting topic, however, the major concerns and inadequate results limit enthusiasm.
Major
1. The conclusion in the abstract “Due to the results of this study, NBL can be used as a therapeutic agent in other lung and tissue injury models with the same mechanism.” It is a very strong statement and can not be concluded from this study alone
2. Immunohistochemistry and gene expression of the markers alone is concerning because the conclusion about the NBL impacts on the leukocyte transendothelial migration would be be strengthened by flow cytometry on the blood and lung tissue.
3. The manuscript requires significant editing. Figure legends must be improved some figures missing labeling.
Minor
1. Result: Suggest substituting “genetic” with gene expression
Examples of inadequate writing include these few examples
2. Abstract: describing the groups “Thirty-two rats were split into four categories: control, LPS (5 mg/kg, intraperitoneally [IP], single dose), LPS (5 mg/kg, I.P, one dosage 30 minutes after last NBL treatment), + NBL (10 mg/kg oral gavage for three days), and NBL. (“and NBL” should be deleted)
3. In the materials and method section on page 2 take out the word materials from this heading “2.1. Animal Material and Experimental Design”
4. on page 3 abbreviations must be defined upon first use
“Negative controls (n = 8): For three days, 1 ml of NS was administered via oral gavage. Thirty minutes after the last SF, one dosage of 1 ml of NS was injected into the right inguinal region of the rat “
Author Response
Please see the attacment

Reviewer 2 Report
In this study, Temel and colleagues evaluate the effect of the β1 adrenoreceptor antagonist nebivolol (NBL) in a model of endotoxin-induced lung injury in the rat. NBL has known anti-inflammatory, antioxidant, and anti-apoptotic properties and has been shown to be protective in hypertension. This study is the first to investigate its use in acute lung injury. The authors first pretreated rats with NBL by oral gavage for 3 days, then administered lipopolysaccharide (LPS) intraperitoneally (IP). Controls included rats given only saline, only NBL, and only LPS. At 6 hours after IP injection, lungs were obtained for analysis. As expected, LPS challenge of rats induced inflammation, apoptosis, and oxidative stress in the lungs, as measured by histology, immunohistochemistry for caspase-3 and ICAM-1, gene expression for MMP-2 and TIMP-1, and biochemical assays for oxidants. All these parameters were reduced by pretreatment with NBL. This is an interesting and novel study showing that blocking an adrenergic pathway dampens inflammation and injury in the lung. However, the mechanism of NBL action on lung inflammation is not known. There are several weaknesses in the study: insufficient background on NBL was provided, limited markers of inflammation were examined, and the immunohistochemical results were not completely convincing. Suggestions for strengthening the manuscript are listed below.
Major:
1. Recommend changing the title to one that is more general, such as “The β1 Adrenergic Blocker Nebivolol Ameliorates Development of Endotoxic Acute Lung Injury.” Although the authors showed that NBL decreases MMP-2 and TIMP-1 mRNA levels, as well as ICAM-1 positivity in tissue sections, these are primarily readouts of inflammation, and other parameters, such as oxidative stress and apoptosis, were examined but not included in the title. There are many factors involved in leukocyte recruitment, and it appears that NBL is a general inhibitor of inflammation.
2. The rationale for focusing primarily on MMP-2/TIMP-1 and ICAM-1 is not completely clear. Because this is the first study to examine the effect of NBL in an acute lung injury model, it would be important to see how gene expression and/or protein of proinflammatory cytokines such as TNF-⍺ are affected, as well as chemokines that are involved in leukocyte recruitment.
3. The immunohistochemical staining, particularly for Cas-3, is not easy to discern in the images for Figs. 3 and 4. The authors should consider adding an inset at high magnification to the images. In Fig. 4, the arrows are not very helpful, as it appears that LPS induces an overall diffuse increase in ICAM-1 positivity, not just at the locations indicated by the arrows.
4. Recommend moving Fig. 1 to the end to serve as a graphical summary of the study results, incorporating the inhibition of these pathways by NBL.
5. Both the Introduction and Discussion should minimize the detailed description of MMPs and TIMPs. Instead, the Introduction should focus more on the background of NBL as a modulator of inflammation in other organs, whereas the Discussion should summarize the current knowledge on potential mechanisms for NBL effects on inflammation and how these might relate to the lung.
6. NBL is being used as a prophylactic in this study, since it is given prior to the onset of inflammation. The authors should speculate on whether it might be equally effective when given therapeutically (i.e., after inflammation has started), as they propose that NBL could be a therapeutic agent. Because NBL has not been tested as therapeutic, recommend modifying the last sentence of the Abstract to something like "The results of this study suggest that NBL has utility as a potential therapeutic agent to dampen inflammation in other lung and tissue injury models."
7. It is intriguing that NBL alone decreases expression of MMP-2 and TIMP-1. It is not clear if the authors considered this in their Discussion.
Minor:
1. Check spelling of all instances of “hyperemic” in Table 2.
2. On page 2, substitute a different word for “increment” in the first paragraph.
3. In paragraph 2 on page 2, does the chemotactic gradient induced by MMP-2 eliminate or recruit inflammatory cells?
4. In section 2.1, were the lungs inflated for fixation?
5. In section 2.1, define the abbreviations “NS” and “SF” when first used.
6. In section 2.3, modify the second paragraph to state “For each section, 10 fields at 400X magnification were evaluated.”
7. In section 2.4, please clarify that 10 mM refers to the concentration of sodium phosphate.
8. In section 2.6 of the Methods, change “results” to “significance.” Modify last sentence to read “Values with p < 0.05 were deemed significant.”
9. The title to section 3.3 should be “Expression results.” In paragraph 2 of this section, “TIMP-1” should be replaced with “MMP-2.”
10. In Figs. 5 and 6, y-axis labels need to be provided. In Fig. 6, please clarify in the legend that expression data were calculated relative to the saline controls.
11. Consider changing some of the terminology when describing the results in Figs. 5 and 6. “Quite a bit” and “a great deal” are not scientifically robust ways to describe data. Also, a 1.4-fold increase would not be considered “a great deal higher,” but moderately increased. In the Discussion, the authors could speculate that these transcriptional changes might be more dramatic at later time points (such as 24 hours after LPS).
12. The data in Figs. 5 and 6 should be shown as individual points in a scatter plot, rather than as columns.
Round 2
Reviewer 1 Report
I am satisfied with the authors' revisions and responses to the comments.
Author Response
Dear referee, thank you for your time and contribution to our study.
Reviewer 2 Report
See attached file.
